# Matching degree of embodied carbon trade and value-added trade among Chinese provinces (regions)

**Xia Li[1], Fengying Lu[2], Guangyao Deng[2]***

**1** School of Economics and Management, Shanghai University of Political Science and Law, Shanghai, PR China, **2** School of Statistics, Lanzhou University of Finance and Economics, Lanzhou, PR China

* dgy316203@163.com

**Data Availability Statement:** Please refer to the following link for the multi-regional input-output table of China in 2012(After registering an account, you can download it for free.) https://www.ceads.net.cn/data/input_output_tables?#1089 Please

## Abstract

This paper constructs a matching index model to research the matching degree of embodied carbon trade and value-added trade among eight regions and 31 provinces in China in 2012 and 2015. The study finds that (1) At the regional level, a mismatch is shown between embodied carbon trade and value-added trade among regions, for example, in 2015, the northwest region has output embodied carbon to the north coast, while input the value-added trades from the north coast. (2) At the provincial level, a mismatch is displayed between embodied carbon transaction and value-added trade, for example, Beijing had a net shift of embodied carbon from Xinjiang in 2015, but Xinjiang had a net transfer of added value from Beijing. Therefore, the Chinese government needs to establish an ecological compensation mechanism to improve the mismatch between embodied carbon trade and value-added trade among Chinese regions (provinces).

## Introduction

As economy develops, China's need for fossil energy is increasing, which the use of it generates large amounts of carbon dioxide. At present, China has surpassed the United States as the top carbon emitter [1, 2]. In response to the biggest threat to humankind, global climate change, the *Sino-US Joint Statement on Climate Change* issued in 2014, stated that "China is scheduled to peak its carbon dioxide emissions around 2030 and will try to realize the peak sooner". In September 2020, General Secretary Xi Jinping pledged at the United Nations General Assembly that "China will strive to realize carbon neutrality by 2060" and in March 2021, Premier Li Keqiang reiterated this goal in his government work report. In order to effectively control carbon emissions and achieve the goals of carbon peaking and carbon neutrality as soon as possible, it is necessary to account for the carbon emissions (i.e., embodied carbon trade) embodied by the transaction of products between regions in China.

In recent years, the connection between production and trade in Chinese provinces (regions) has become closer and closer. Value-added trade is the added trade produced in different provinces (regions) by different production steps since the production of a product is mostly done by many enterprises in multiple provinces (regions) [3–6]. By combining

refer to the following link for the multi-regional input-output table of China in 2015(After registering an account, you can download it for free.) https://www.ceads.net.cn/data/input_output_tables?#926 Please refer to the following link for the carbon emission data of Chinese provinces in 2012(After registering an account, you can download it for free.) https://www.ceads.net.cn/data/province/by_sectoral_accounting/Provincial?#970 Please refer to the following link for the carbon emission data of Chinese provinces in 2015 (After registering an account, you can download it for free.) https://www.ceads.net.cn/data/province/by_sectoral_accounting/Provincial?#973.

**Funding:** This work was supported by the Natural Science Foundation of China under Grant#71704070(GD); Outstanding Youth Fund of Gansu Province#20JR5RA206(GD); Gansu Provincial Higher Education Research Project#2020A-058(GD); Double First-class Key Scientific Research Project of Gansu Provincial Department of Education # GSSYLXM-06(GD); Key scientific research project of Silk Road Economic Research Institute of Lanzhou University of Finance and Economics # JYYZ202102(GD); Program of Lanzhou University of Finance and Economics under Grant#Lzufe2021B-002(GD).The recipients of the above funds are Guangyao Deng, the corresponding author of this article. The funders had no role in study design, data collection and analysis, decision to publish, or preparation of the manuscript.

**Competing interests:** The authors have declared that no competing interests exist.

the value-added trade and the embodied carbon trade, some scholars have also proposed the concept of aggregate embodied intensity (AEI) and applied it to the related research of China's embodied carbon trade [7–10]. The AEI indicator can be defined at the aggregation, regional, final demand, sectoral and transmission layers. It should be noted that the rewards brought by the value-added trade and the embodied carbon trade to different regions are not uniform. In most cases, the transfer in place of the value-added trade should pay an appropriate amount of remuneration for the products of the transfer out place of value-added trade. However, for the embodied carbon trade, there is a lack of corresponding eco-logical compensation mechanism. When a region transfers products with the high carbon emissions from other regions to replace the local production, the ecological environment of region is not damaged and the environmental cost is not paid. However, other regions need to bear the environmental pollution caused by the high carbon emissions, which leads to the so-called carbon leakage phenomenon [2]. In order to better clarify the carbon emission reduction responsibilities of provinces (regions), it is necessary to analyze the degree of matching between embodied carbon trade and value-added trade by combining the calcula-tion results of them.

Currently, the input-output model is mainly used to study embodied carbon trade. According to the number of regions involved in the input-output model, it can be fallen into single-region input-output model and multi-region input-output model. (1) In terms of single-region input-output models. Ren et al. [11] assessed the embodied carbon trade of 19 industries in China from 2001 to 2011, and pointed out that the embodied carbon trans-action in China showed a continuous growth trend. Su and Thomson [12] calculated the embodied carbon trade of 135 industries in China from 2006 to 2012, and clarified that the embodied carbon emissions of normal exports and processed exports grew from 2006 to 2008, and then during the global financial crisis (2008–2009) decreased and increased again after 2009.Liu et al. [13] studied the embodied carbon emissions of Liaoning province in China in 2012, noting that the growth of product exports and inter-provincial transfers were important factors leading to the growth of carbon emissions. (2) In terms of multi-regional input-output models. Cheng et al. [14] studied the embodied carbon trade among provinces in the middle reaches of the Yangtze River region in China from 2007 to 2012, pointing out that inter-area embodied carbon emissions flowed to relatively developed areas including Jiangsu and Shanghai. Wang et al. [15] studied the embodied carbon trade among eight areas in China in 2007 and 2012, pointing out that meeting domestic demand is the main source of productive carbon emissions in China. However, the share of carbon emissions from external demand is increasing rapidly. Liu et al. [16] researched the embod-ied carbon trade between China and the U.S. in 2017, noting that the trade between them resulted in a net shift of 242 Mt $CO_2$ per year from the U.S. to China. Deng et al. [6] calcu-late the amount of embodied carbon trade of China from 2006 to 2015, pointing out that the embodied carbon trade between China and the United States is large. The above literatures have made many studies on the embodied carbon trade in China as well as other countries, but still have not analyzed the degree of matching between embodied carbon trade and value added trade among regions (provinces) in China.

The main work done in this paper is as follows: (1) The multi-regional input-output model was used for the calculation of the embodied carbon trade and value-added trade between China's eight regions and 31 provinces in 2012 and 2015. (2) Analyzed the matching degree of embodied carbon trade and value-added trade among various regions (provinces) in China.

## Methods

### Accounting for inter-regional embodied carbon trade

On basis of the structure of China's interregional input-output tables [17]:

$$
\begin{bmatrix} X^1 \\ \vdots \\ X^n \end{bmatrix} = \begin{bmatrix} A^{11} & \cdots & A^{1n} \\ \vdots & \ddots & \vdots \\ A^{n1} & \cdots & A^{nn} \end{bmatrix} \begin{bmatrix} X^1 \\ \vdots \\ X^n \end{bmatrix} + \begin{bmatrix} f^{11} & \cdots & f^{1n} & f^{1,row} \\ \vdots & \ddots & \vdots & \vdots \\ f^{n1} & \cdots & f^{nn} & f^{n,row} \end{bmatrix} e \tag{1}
$$

Wherein, $X^1$ is the overall output column vector of region 1, $A^{11}$ is the direct consumption index matrix of region 1 for the intermediate use of its own products, $A^{n1}$ means the direct consumption index matrix of area 1 for the intermediate use of products from region n, $f^{11}$ is the final use column vector of region 1 for its own products, $f^{1n}$ is the transfer out of region 1 to region n, $f^{1,row}$ is the export of region 1, $e$ is the unit row vector, other symbols are similarly defined. Formula (1) can be deformed to:

$$
\begin{bmatrix} X^1 \\ \vdots \\ X^n \end{bmatrix} = \left\{ \begin{bmatrix} I & \cdots & 0 \\ \vdots & \ddots & \vdots \\ 0 & \cdots & I \end{bmatrix} - \begin{bmatrix} A^{11} & \cdots & A^{1n} \\ \vdots & \ddots & \vdots \\ A^{n1} & \cdots & A^{nn} \end{bmatrix} \right\}^{-1} \begin{bmatrix} f^{11} & \cdots & f^{1n} & f^{1,row} \\ \vdots & \ddots & \vdots & \vdots \\ f^{n1} & \cdots & f^{nn} & f^{n,row} \end{bmatrix} e = \begin{bmatrix} L^{11} & \cdots & L^{1n} \\ \vdots & \ddots & \vdots \\ L^{n1} & \cdots & L^{nn} \end{bmatrix} \begin{bmatrix} f^{11} & \cdots & f^{1n} & f^{1,row} \\ \vdots & \ddots & \vdots & \vdots \\ f^{n1} & \cdots & f^{nn} & f^{n,row} \end{bmatrix} e (2)
$$

Among them, $I$ is the unit matrix. This paper further defines the direct carbon emission factor according to the following Eq:

$$
c_i^r = \frac{CO_{2,i}^r}{X_i^r} \tag{3}
$$

Among them, $CO_{2,i}^r$, $X_i^r$ respectively show the $CO_2$ emissions and total output (total input) of region r sector i. Referring to Duarte et al. [18], the embodied trade volume between regions in China can be calculated by the following Eq:

$$
D = CLF \tag{4}
$$

Among them, $C$ is the diagonal matrix formed by the direct carbon emission factors, $L$ means the Leontief inverse matrix, and $F$ refers to the end-use matrix, the specific equation of the end-use matrix is as follows:

$$
F = \begin{bmatrix} f^{11} & \cdots & f^{1n} \\ \vdots & \ddots & \vdots \\ f^{n1} & \cdots & f^{nn} \end{bmatrix} \tag{5}
$$

The calculation results in Eq (4) are collated to obtain the embodied carbon trade between the 31 provinces in mainland China and between the eight regions. The net embodied carbon transfer between region r and s is calculated according to the following Eq:

$$
\bar{D}^{rs} = D^{rs} - D^{sr} \tag{6}
$$

## Calculation of inter-regional value-added trade

Similar to the calculation of embodied carbon trade, this paper first defines the value-added coefficient:

$$v_i^r = \frac{V_i^r}{X_i^r} \tag{7}$$

$V_i^r$, $X_i^r$ respectively represent the added value and total input of region r sector i. The value-added trade among regions can be calculated by the following formula:

$$W = VLF \tag{8}$$

V refers to the diagonal matrix generated by the value added coefficients, and the names of other variables are consistent with Eq (4). Similarly, the net transfer out of the value added between areas r and s is calculated according to the following Eq:

$$\bar{W}^{rs} = W^{rs} - W^{sr} \tag{9}$$

## Matching index of embodied carbon trade and value-added trade

Since the net embodied carbon transfer can be positive or negative, this paper refers to Zhang et al. [19], Chen et al. [20] and converts the net embodied carbon transfer to a non-negative number by the following Eq:

$$\bar{DN}^{rs} = \frac{(\bar{D}^{rs} + |\bar{D}^{rs}|)}{2} \tag{10}$$

|| denotes the absolute value. Further, the matching degree of embodied carbon trade and value-added trade is calculated according to the following Eq:

$$CI^{rs} = \begin{cases} f(\frac{\bar{DN}^{rs}}{\bar{W}^{rs}}), if \bar{DN}^{rs} > 0 \, and \, \bar{W}^{rs} > 0 \\ f(\bar{DN}^{rs}) + f(|\bar{W}^{rs}|) + 1, if \bar{DN}^{rs} > 0 \, and \, \bar{W}^{rs} < 0 \\ 0, if \bar{DN}^{rs} = 0 \, or \, \bar{W}^{rs} = 0 \end{cases} \tag{11}$$

where f(y) is defined by the following Eq:

$$f(y) = \frac{y - y_{min}}{y_{max} - y_{min}} \tag{12}$$

In the above equation, $y_{max}$ and $y_{min}$ respectively represent the maximum and minimum values in a set of data. When $\bar{DN}^{rs} > 0$, it indicates that the net transfer out of embodied carbon from region $r$ to region $s$ is greater than 0, and both $\bar{w}^{rs} > 0$ and $\bar{w}^{rs} < 0$ may exist. If $\bar{DN}^{rs} > 0$ and $\bar{w}^{rs} > 0$, it implies that the net transfer out of both embodied carbon and added value from region $r$ to region $s$ is greater than 0. The representative element $\frac{\bar{DN}^{rs}}{\bar{w}^{rs}}$ in the matrix can be normalized to 0–1 by the function $f(y)$. The larger the net transfer out of embodied carbon from region $r$ to region $s$, and the smaller the net transfer out of added value, the closer $CI^{rs}$ is to 1. If $\bar{DN}^{rs} > 0$ and $\bar{w}^{rs} < 0$, the net transfer out of embodied carbon from region $r$ to region $s$ is greater than 0, but the net transfer out of added value is less than 0. In other words, the added value of region $r$ is net transfer in from region $s$, which reflects the mismatch between the embodied carbon trade and the added-value trade. By Eq(11), the net transfer out of embodied carbon and the net transfer in of added value from region $r$ to region $s$ can be normalized to 0–1, and to distinguish from the first case, 1 is added on this basis. The larger the

net transfer out of embodied carbon from region r to region s, and the higher the net transfer in of added value from region r to region s, the greater $CI^{rs}$ is greater than 1.It should be noted that when $\bar{DN}^{rs} = 0$, the net embodied carbon transfer from area r to area s is less than 0, i.e., there is a net embodied carbon transfer from area r to area s. In addition, according to Eq (11), $CI^{rs} \neq CI^{s}$, that is, the inequality of carbon emissions in the direction of net shift out and net shift in of each region is distinguished.

## Data

Due to the tedious process of compiling interregional input-output tables, the latest interregional input-output tables for China correspond to the years 2012 and 2015. Among them, the 2012 interregional input-output table is from Liu et al. [17], and the 2015 interregional input-output table is from the CEADs database [21]. The carbon emission data of each province and industry in China are calculated according to the carbon emission accounting inventory method proposed by IPCC (2006).Since China's multi-regional input-output table in CEADs database includes Tibet, but lacks its energy and carbon emission data of various industries, this research assumes that the direct carbon emission coefficient of various industries in Tibet is equal to the average value of other 30 provinces, which is used to estimate the carbon emission of Tibet. In addition, because Tibet has less trade links with other provinces, there is less embodied carbon trade with other provinces. Based on the above viewpoint, the treatment of carbon emissions in Tibet will not cause a large deviation to the results. In consideration of the comparability of data in different years, this study takes 2012 as the base period and deflates the data related to prices in 2015.In addition, this paper divides the 31 provinces in mainland China into eight regions with reference to *the Strategies and Policies for Coordinated Regional Development* published by the Growth Research Center of the State Council of China as shown in Table 1.

## Results and discussions

### Inter-regional embodied carbon trade

The calculations of embodied carbon trade among the eight regions in China in 2015 are displayed in Table 2.

Each row of data in Table 2 indicates the amount of embodied carbon trade shifted out from each region to other regions, and each column of data indicates the amount of embodied carbon trade shifted in from other areas by each region, for example, the amount of embodied carbon trade of commodities transferred out from the northeast to the northern coastal area in 2015 is 21,697,100 tons, which means the amount of embodied carbon transaction of products transferred in from the northeast region by the northern coastal area is 21,697,100 tons. Since the elements on the diagonal are actually the embodied carbon consumption of every area itself, the diagonal elements are recorded as 0 in this paper.

Comparing the results of embodied carbon transfer out and in from each region, it can be found that (1) the middle Yellow River region transferred the greatest amount of embodied

**Table 1. Eight regions and 31 provinces.**

| Region | Province | Region | Province |
|---|---|---|---|
| northeast | Liaoning, Jilin, Heilongjiang | middle Yellow River | Shaanxi, Shanxi, Henan, Inner Mongolia |
| northern coastal | Tianjin, Beijing, Hebei, Shandong | middle Yangtze River | Hubei, Hunan, Jiangxi, Anhui |
| eastern coastal | Shanghai, Jiangsu, Zhejiang | southwest | Yunnan, Guizhou, Sichuan, Chongqing, Guangxi |
| southern coastal | Fujian, Guangdong, and Hainan | northwest | Gansu, Qinghai, Ningxia, Tibet, Xinjiang |

**Table 2. Embodied carbon trade among eight regions in China in 2015 (unit: 10,000 tons).**

| Region | Northeast | North coast | East coast | South coast | Middle Yellow River | Middle Yangtze River | South west | Northwest | Total transfer out |
|---|---|---|---|---|---|---|---|---|---|
| Northeast | 0.00 | 2169.71 | 2591.87 | 1815.11 | 1410.39 | 2250.79 | 1832.32 | 658.20 | 12728.40 |
| North coast | 1364.72 | 0.00 | 3393.76 | 2695.87 | 1965.71 | 2658.02 | 2639.63 | 717.24 | 15434.95 |
| East coast | 895.16 | 2025.36 | 0.00 | 1714.11 | 1408.40 | 2094.18 | 2043.54 | 547.79 | 10728.55 |
| South coast | 440.49 | 986.65 | 1045.41 | 0.00 | 649.55 | 1312.65 | 1054.66 | 249.83 | 5739.24 |
| Middle Yellow River | 1762.43 | 4356.63 | 5213.11 | 3235.09 | 0.00 | 5633.94 | 3444.74 | 804.42 | 24450.36 |
| Middle Yangtze River | 615.56 | 1291.50 | 2011.78 | 1646.73 | 1101.32 | 0.00 | 1518.59 | 377.37 | 8562.83 |
| Southwest | 521.49 | 1153.08 | 1664.82 | 1456.75 | 1074.38 | 1538.30 | 0.00 | 509.96 | 7918.79 |
| Northwest | 598.50 | 1840.56 | 1909.16 | 1361.00 | 1259.49 | 2357.96 | 1455.53 | 0.00 | 10782.20 |
| Total transfer in | 6198.37 | 13823.48 | 17829.92 | 13924.65 | 8869.25 | 17845.84 | 13989.01 | 3864.81 | 96345.32 |

carbon emissions to other regions in 2015, 244,503,600 tons; the southern coastal region transferred the smallest amount of embodied carbon emissions to other regions, 57,392,400 tons. (2) The middle Yangtze River region had the largest embodied carbon emissions from other areas in 2015, 178,458,400 tons; the northwest region had the smallest embodied carbon emissions from other areas, 3,864,800 tons. (3) The total embodied carbon emissions transferred out and transferred in from the middle Yellow River region in 2015 was the largest, 333,190,000 tons (24450.36+8869.25, there is a rounding mistake of 0.01 million tons); the total embodied carbon emissions transferred out and transferred in from the northwest region was the smallest, 146,470,100 tons (10,782.20+3864.81). In addition, because domestic trade is a closed whole, the total value of embodied carbon transfer out from the eight regions is equivalent to the total value of embodied carbon transfer in, both of which is 963,453,200 tons.

According to the calculation Eq (4) of embodied carbon trade, it can be seen that the amount of embodied carbon trade is mainly determined by the trade volume of the final use product., In addition, there are the effects of direct carbon emission index and Leontief inverse matrix (which can be interpreted as the joint influence of both industrial structure and trade in intermediate use part of products). Furthermore, the amount of embodied carbon trade between regions is related to factors such as the traffic conditions between the regions, the geographical distance and other factors. The better the traffic conditions between the two regions, the closer the geographical location. The embodied energy carbon trade is more likely to be larger. Based on the classical trade gravity model, the trade volume between the two regions is directly proportional to the total economic volume and inversely proportional to the trade distance [22].The data in this paper also support the above conclusion. For example, the northern and eastern coastal areas are economically developed areas in China with high economic aggregate and adjacent geographical location, which results in relatively higher the volume of embodied carbon trade between these two areas. While, the economic aggregate of Northwest China is lower and the geographical location is far from the northern coastal area, which exhibits a relatively lower the embodied carbon trade between the two regions. The calculation outcomes of the embodied carbon trade among the eight regions in China in 2012 are shown in Fig 1.

The width of the outer arc is in Fig 1 is used to assess the sum of the embodied carbon transfer out and transfer in between each region and the other seven regions, the width of the inner arc is used to measure the sum of the embodied carbon transfer out from each region to the other seven regions, and the width corresponding to the difference between the outer arc and the inner arc is used to assess the sum of the embodied carbon transfer in from each region to the other seven regions. It can be seen from Fig 1 that the outer arc of the middle Yellow River

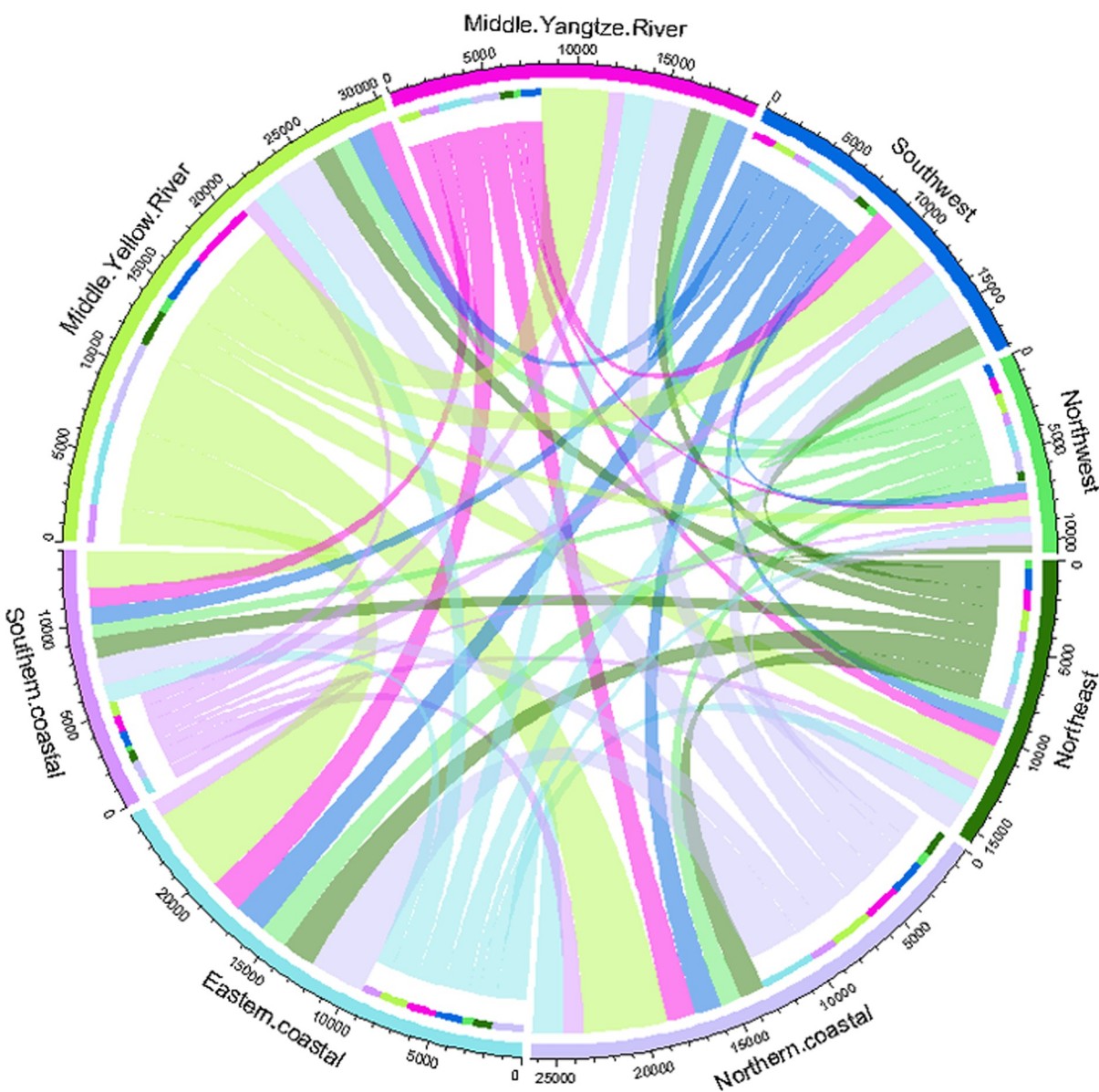

**Fig 1. Embodied carbon trade among the eight regions in 2012 (unit: 10,000 tons).**

region is the widest, which indicates that the sum of the embodied carbon shift out and shift in between the middle Yellow River region and the other seven regions is the largest; the outer arc of the northwest region is the narrowest, which indicates that the sum of the embodied carbon transfer out and transfer in between the northwest region and the other seven areas is the smallest. In addition, the width of the inner arc in the middle Yellow River region is larger than the width corresponding to the difference between the outer arc and the inner arc, which indicates that the embodied carbon transfer out of the Yellow River basin is larger than the embodied carbon transfer in, that is, there is a net embodied carbon transfer out of the middle Yellow River region in 2012.

The width of the connecting line between the two regions in Fig 1 is used to measure the magnitude of the embodied carbon trade between the two regions, and the colors of the

connecting lines are not the same for different regional departures (the direction of embodied carbon transfer out). From Fig 1, we can see that: the arcs of the middle Yellow River region transferred out to the eastern coastal area, the northern coastal area and the middle Yangtze River region are wider, which means that the embodied carbon trade is relatively larger in the middle Yellow River region transferred out to the eastern coastal area, the northern coastal area and the middle Yangtze River region, which is basically consistent with the results of 2015 in Table 2.

This paper further analyzes the embodied carbon trade of 31 provinces in China, and the calculation outcomes of the embodied carbon transaction of each province in 2012 and 2015 are shown in Table 3 (the embodied carbon trade between two of the 31 provinces forms a trade matrix of order 31 × 31, and only the total embodied carbon emissions shifted out of each province to other provinces and the total embodied carbon emissions shifted in from other provinces are listed in Table 3).

According to Table 3: (1) the total value of embodied carbon transfer out from 31 provinces in the same year is equivalent to the total value of embodied carbon transfer in, which was 890,394,700 tons and 105,290,750 tons in 2012 and 2015, respectively. This is because domestic trade is a closed one, and the embodied carbon transfer out from province A to province B is equal to the embodied carbon transfer in from province A by province B. (2) Compared with 2012, the amount of embodied carbon transfer out and transfer in of most provinces in 2015 has increased to different degrees. This is due to the convenience of transportation and the increasing trade volume of products, which leads to the increase in embodied carbon trade. However, it should be pointed out that the direct carbon emission factors of most industries in the provinces decreased in 2015 due to the improvement of energy-saving and emission-decrease technologies, which will lead to a decrease in embodied carbon trade. Under the combined effect of both, the embodied carbon transfers out and transfers in decreased in some provinces. (3) In both 2012 and 2015, the net embodied carbon transfers out were larger in Inner Mongolia and Shanxi, because Inner Mongolia and Shanxi are China's major energy provinces, and more products from the electricity, gas and water production and supply industries in these two provinces are transferred out to other provinces, and the direct carbon emission factors of the electricity, gas and water production and supply industries are larger than those of other industries. (4) In 2012 and 2015, compared with other provinces, Guangdong Province had the largest net embodied energy transfer in, with values of 31,096,300 tons and 93,975,000 tons, respectively. This is because of the shortage of energy in Guangdong Province, and a large number of manufacturing and supply products of electricity, gas and water need to be transferred from other provinces.

### Inter-regional in value-added trade

The calculation results of value-added trade among the eight regions in China in 2015 are shown in Table 4.

Comparing the results of value added transfer out and transfer in of each region, it can be found that: both value added transfer out and value added transfer in were the largest in the eastern coastal region, with 1,431.010 billion yuan and 1,252.647 billion yuan, respectively, and the total value added transfer out and transfer in was 2,683.657 billion yuan; the smallest in the northwest region, which was 332.704 billion yuan and 404.018 billion yuan, respectively. The total value added transferred out and transferred in was 736.722 billion yuan. In addition, the total value of value added transferred out from each region was equivalent to the total value of value added transferred in, which was 736.722 billion yuan. The results in Table 4 also support the conclusion in the classical trade gravity model [22]. For instance, the geographically

Table 3. Embodied carbon trade by province in China in 2012 and 2015 (unit: 10,000 tons).

| Province | 2012 | | | 2015 | | |
|---|---|---|---|---|---|---|
| | Transfer out | Transfer in | Net transfer out | Transfer out | Transfer in | Net transfer out |
| Beijing | 1927.30 | 3432.83 | -1505.53 | 1360.24 | 7341.05 | -5980.81 |
| Tianjin | 1957.87 | 2374.94 | -417.06 | 1702.32 | 2208.87 | -506.56 |
| Hebei | 5448.68 | 5106.71 | 341.97 | 7362.60 | 2406.51 | 4956.09 |
| Shanxi | 7096.37 | 2254.01 | 4842.36 | 6913.81 | 1453.32 | 5460.49 |
| Inner Mongolia | 8180.55 | 2136.81 | 6043.74 | 10278.45 | 1149.46 | 9128.98 |
| Liaoning | 2898.34 | 3342.77 | -444.44 | 4732.00 | 2283.28 | 2448.73 |
| Jilin | 2615.63 | 2109.71 | 505.92 | 3331.17 | 2028.18 | 1302.99 |
| Heilongjiang | 3449.54 | 2490.89 | 958.65 | 5379.83 | 2601.52 | 2778.32 |
| Shanghai | 2730.20 | 5043.80 | -2313.60 | 2392.77 | 5491.37 | -3098.60 |
| Jiangsu | 4873.02 | 6267.40 | -1394.38 | 6818.86 | 6335.48 | 483.38 |
| Zhejiang | 3135.68 | 5977.32 | -2841.64 | 3110.57 | 7596.71 | -4486.15 |
| Anhui | 4320.58 | 3637.53 | 683.04 | 4749.20 | 5288.66 | -539.46 |
| Fujian | 1731.08 | 1584.15 | 146.93 | 2629.77 | 1534.30 | 1095.47 |
| Jiangxi | 1137.85 | 3718.62 | -2580.77 | 1598.21 | 3677.31 | -2079.09 |
| Shandong | 6227.14 | 4619.57 | 1607.57 | 7252.58 | 4109.83 | 3142.74 |
| Henan | 4525.21 | 3816.24 | 708.97 | 5223.00 | 4533.70 | 689.31 |
| Hubei | 1828.83 | 3150.30 | -1321.47 | 1081.34 | 6878.93 | -5797.59 |
| Hunan | 1867.34 | 2350.83 | -483.49 | 2127.19 | 3040.75 | -913.57 |
| Guangdong | 3451.74 | 6561.37 | -3109.63 | 2789.16 | 12186.66 | -9397.50 |
| Guangxi | 1265.97 | 2017.69 | -751.72 | 1490.96 | 2197.92 | -706.96 |
| Hainan | 658.68 | 641.76 | 16.92 | 768.49 | 605.16 | 163.32 |
| Chongqing | 1361.28 | 2872.86 | -1511.58 | 1255.18 | 5296.00 | -4040.82 |
| Sichuan | 1486.57 | 2327.34 | -840.78 | 1765.22 | 2586.58 | -821.36 |
| Guizhou | 2755.58 | 1272.66 | 1482.91 | 2732.68 | 1845.86 | 886.82 |
| Yunnan | 1826.34 | 2482.10 | -655.76 | 1519.10 | 3328.73 | -1809.64 |
| Xizang | 25.07 | 253.41 | -228.34 | 31.63 | 83.43 | -51.81 |
| Shanxi | 3536.71 | 3522.21 | 14.50 | 4128.60 | 3402.52 | 726.08 |
| Gansu | 1770.49 | 1246.45 | 524.04 | 1868.16 | 1293.63 | 574.53 |
| Qinghai | 416.45 | 343.15 | 73.30 | 463.71 | 518.08 | -54.37 |
| Ningxia | 1889.67 | 474.43 | 1415.24 | 2015.52 | 314.62 | 1700.89 |
| Xinjiang | 2643.73 | 1609.60 | 1034.13 | 6418.43 | 1672.30 | 4746.13 |
| Total | 89039.47 | 89039.47 | 0.00 | 105290.75 | 105290.75 | 0.00 |

Table 4. Value-added trade among eight regions in China in 2015 (unit: billion yuan).

| Region | Northeast | North coast | East coast | South coast | Middle Yellow River | Middle Yangtze River | Southwest | Northwest | Total transfer out |
|---|---|---|---|---|---|---|---|---|---|
| Northeast | 0.00 | 1578.77 | 1917.98 | 1569.16 | 1141.75 | 1351.34 | 1438.37 | 666.29 | 9663.66 |
| North coast | 1244.98 | 0.00 | 2731.43 | 2465.23 | 1721.04 | 2392.58 | 2311.52 | 689.31 | 13556.10 |
| East coast | 1087.91 | 2824.75 | 0.00 | 2295.09 | 1922.26 | 2590.90 | 2786.58 | 802.62 | 14310.10 |
| South coast | 575.72 | 1114.81 | 1149.89 | 0.00 | 875.13 | 1090.14 | 1365.71 | 358.45 | 6529.84 |
| Middle Yellow River | 890.17 | 1827.00 | 2582.15 | 1903.65 | 0.00 | 2049.06 | 1846.31 | 520.76 | 11619.11 |
| Middle Yangtze River | 634.08 | 1437.89 | 1869.42 | 1828.44 | 1348.76 | 0.00 | 1758.53 | 431.79 | 9308.91 |
| Southwest | 517.18 | 1279.23 | 1661.19 | 1667.39 | 1300.89 | 1286.13 | 0.00 | 570.96 | 8282.98 |
| Northwest | 180.24 | 522.94 | 614.40 | 595.96 | 516.27 | 464.11 | 433.11 | 0.00 | 3327.04 |
| Total transfer in | 5130.28 | 10585.40 | 12526.47 | 12324.92 | 8826.09 | 11224.26 | 11940.13 | 4040.18 | 76597.73 |

adjacent northern and eastern coastal areas in China are developed areas with a relatively high economic aggregate, and the volume of added value trade between these two areas is relatively large. While, the Northwest China, which is far from the northern coastal areas, has a relatively low economic aggregate and the volume of added value trade between the two regions is also comparatively low.

Fig 2 shows the calculation results of value-added trade among the eight areas in China in 2012.

The meaning of each graphic element in Fig 2 is consistent with Fig 1. From Fig 2, we can see that: (1) the width of the outer arc corresponding to the northern coastal area, the eastern coastal area and the middle Yellow River area is wider, which indicates that the value-added trade between the northern coastal area, the eastern coastal area and the middle Yellow River region and other regions is relatively larger (including transfer out and transfer in). (2) The width of the inner arc in the northern coastal region is the widest, and the width of the inner arc in the northwest region is the narrowest, which indicates that the amount of value added transferred out of the northern coastal region to other areas is increased, while the amount of value added transferred out of the northwest region to other areas is smaller. (3) The width of the connecting line from the northern coastal region to the eastern coastal region as well as the middle Yellow River region, and the width of the connecting line from the middle Yellow River region to the eastern coastal region is wider, which means that the volume of value added trade from the northern coastal area to the eastern coastal area as well as the middle Yellow River area, and the volume of value added trade from the middle Yellow River area to the eastern coastal area is relatively larger.

This paper further analyzes the value-added trade of 31 provinces in China, and the calculation results of the value-added trade of each province in 2012 and 2015 are shown in Table 5 (the value-added trade between two of the 31 provinces forms a trade matrix of order $31 \times 31$, and only the total value-added trade transferred out of each province to other provinces and the total value-added trade transferred in from other provinces are shown in Table 5).

It can be seen from Table 5: (1) Similar to the outcomes in Table 3, the total value of added value transfers out of 31 provinces in the same year is equivalent to the total value of added value transfers in, which are 6,025,524 million yuan and 8,431,006 million yuan in 2012 and 2015, respectively. Compared with 2012, the added value transfers in and out of most provinces in 2015 have increased to different degrees. (2) Compared with other provinces, Guangdong Province has the largest net value added transfer in, with values of 190.188 billion yuan and 717.067 billion yuan in 2012 and 2015, respectively. This is due to the fact that Guangdong Province is the most populous province in China, which needs to transfer a large number of agricultural products from other regions, while the higher labor input and larger value added coefficient in agriculture, together with the effect of value added trade in other industries, lead to a larger net value added transfer in the province. (3) Comparing the results in Table 3, there are some differences between the calculation results of embodied carbon trade and the calculation results of value-added trade in each province. Some provinces have net embodied carbon transfer in, but there is net value added transfer out in the same year, for example, Beijing had net embodied carbon transfer in (150.553 million tons) but there was a net value added transfer out (126.832 billion yuan) in 2012. The reason for this phenomenon is that industries with higher (lower) value-added coefficients may not necessarily have higher (lower) direct carbon emission coefficients.

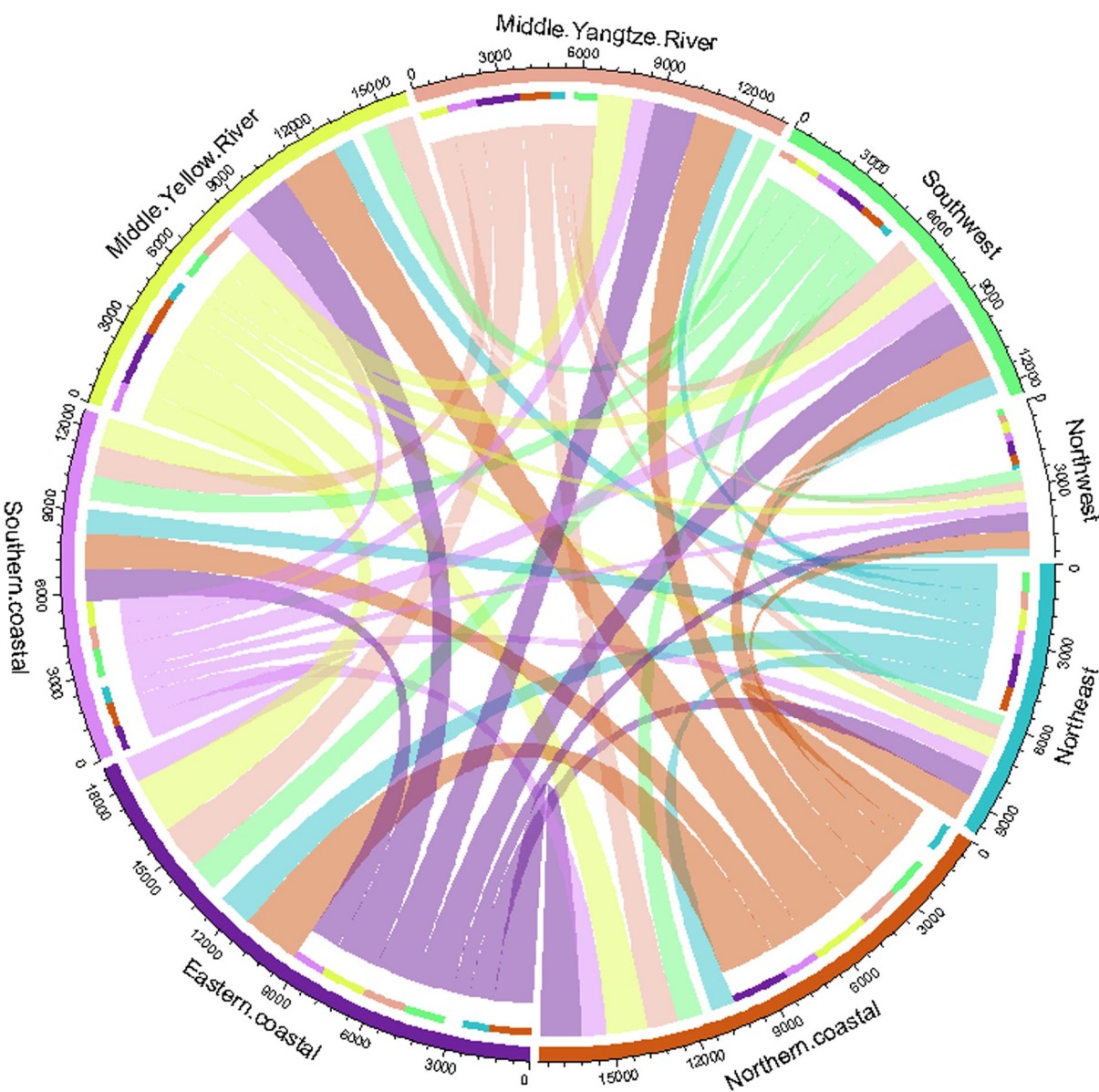

**Fig 2. Value-added trade among eight regions in China in 2012 (unit: billion yuan).**

## Matching degree of embodied carbon trade and value-added trade in each region

In this paper, we calculate the matching degree of embodied carbon trade and value-added trade between the eight regions in 2012 and 2015 based on Eq (11), The specific results are shown in Tables 6 and 7.

From Tables 6 and 7, we can see that: (1) the elements on the diagonal are all 0. This is because the embodied carbon trade and value-added trade do not occur between the regions themselves and themselves. (2) The values of the direction of transfer from the northern coastal area to the northeast in Table 5 and the direction of transfer from the east-central coast to the middle reaches of the Yangtze River in Table 6 are both 0, but marked with $^*$. The $^*$ sign indicates the minimum value in the set of data calculated according to Eq (11) $\frac{DN^{rs}}{W^{rs}}$. (3) In

**Table 5. Value-added trade volume of each province in 2012 and 2015 (unit: billion yuan).**

| Province | 2012 | | | 2015 | | |
|---|---|---|---|---|---|---|
| | Transfer out | Transfer in | Net transfer out | Transfer out | Transfer in | Net transfer out |
| Beijing | 3789.26 | 2520.94 | 1268.32 | 2835.38 | 4774.21 | -1938.83 |
| Tianjin | 2030.91 | 1121.79 | 909.12 | 2319.95 | 1889.91 | 430.04 |
| Hebei | 2456.77 | 2548.85 | -92.07 | 4422.57 | 2290.96 | 2131.61 |
| Shanxi | 1622.20 | 1920.12 | -297.93 | 1744.73 | 1314.67 | 430.06 |
| Inner Mongolia | 2132.12 | 1936.54 | 195.58 | 2753.70 | 1309.28 | 1444.42 |
| Liaoning | 1816.30 | 1944.33 | -128.03 | 3749.23 | 1896.99 | 1852.23 |
| Jilin | 1587.80 | 1093.09 | 494.72 | 2750.89 | 1284.98 | 1465.91 |
| Heilongjiang | 2456.20 | 1799.63 | 656.57 | 3607.11 | 2391.88 | 1215.23 |
| Shanghai | 3307.35 | 3458.11 | -150.76 | 3619.87 | 4397.07 | -777.20 |
| Jiangsu | 4904.66 | 4115.03 | 789.63 | 9391.87 | 4377.24 | 5014.64 |
| Zhejiang | 3000.18 | 3989.40 | -989.22 | 3512.90 | 5966.70 | -2453.80 |
| Anhui | 2653.48 | 2669.41 | -15.93 | 3523.79 | 5280.57 | -1756.77 |
| Fujian | 1878.41 | 1337.69 | 540.72 | 2924.04 | 1816.29 | 1107.75 |
| Jiangxi | 1227.74 | 1691.32 | -463.58 | 1871.26 | 1877.83 | -6.57 |
| Shandong | 4224.62 | 3258.45 | 966.17 | 5782.69 | 3434.81 | 2347.87 |
| Henan | 2997.38 | 3190.58 | -193.20 | 4971.29 | 4094.50 | 876.79 |
| Hubei | 1248.44 | 1490.93 | -242.50 | 1436.47 | 3145.13 | -1708.65 |
| Hunan | 2197.33 | 1683.57 | 513.76 | 3463.25 | 1950.54 | 1512.70 |
| Guangdong | 3483.67 | 5385.56 | -1901.88 | 3288.14 | 10458.81 | -7170.67 |
| Guangxi | 1369.50 | 1484.28 | -114.77 | 1877.43 | 1912.06 | -34.63 |
| Hainan | 718.43 | 563.49 | 154.94 | 951.99 | 640.21 | 311.79 |
| Chongqing | 1140.41 | 1548.05 | -407.64 | 1612.62 | 4559.56 | -2946.93 |
| Sichuan | 1370.53 | 1728.96 | -358.43 | 1995.90 | 2271.63 | -275.73 |
| Guizhou | 916.73 | 1087.65 | -170.92 | 1774.64 | 1755.93 | 18.71 |
| Yunnan | 1106.36 | 1822.56 | -716.20 | 1721.46 | 2658.65 | -937.19 |
| Xizang | 61.26 | 168.98 | -107.73 | 83.83 | 91.03 | -7.21 |
| Shanxi | 2331.24 | 1725.07 | 606.17 | 3196.82 | 2683.07 | 513.75 |
| Gansu | 691.37 | 952.32 | -260.96 | 1151.45 | 1166.02 | -14.58 |
| Qinghai | 177.89 | 296.22 | -118.33 | 185.39 | 450.75 | -265.36 |
| Ningxia | 267.53 | 376.92 | -109.38 | 284.09 | 289.74 | -5.65 |
| Xinjiang | 1089.16 | 1345.39 | -256.23 | 1505.30 | 1879.04 | -373.74 |
| Total | 60255.24 | 60255.24 | 0.00 | 84310.06 | 84310.06 | 0.00 |

**Table 6. The matching degree of embodied carbon trade and value-added trade among the eight regions in 2012.**

| Region | Northeast | North coast | East coast | South coast | Middle Yellow River | Middle Yangtze River | Southwest | Northwest |
|---|---|---|---|---|---|---|---|---|
| Northeast | 0.0000 | 0.0000 | 0.1398 | 0.0877 | 0.0000 | 0.1432 | 0.0595 | 0.0000 |
| North coast | 0.0000* | 0.0000 | 0.1152 | 0.0483 | 0.0000 | 0.0598 | 0.0373 | 0.0000 |
| East coast | 0.0000 | 0.0000 | 0.0000 | 0.0150 | 0.0000 | 0.0000 | 0.0000 | 0.0000 |
| South coast | 0.0000 | 0.0000 | 0.0000 | 0.0000 | 0.0000 | 0.0000 | 0.0000 | 0.0000 |
| Middle Yellow River | 1.3907 | 2.7451 | 0.4513 | 0.4246 | 0.0000 | 0.5196 | 1.0000 | 0.0000 |
| Middle Yangtze River | 0.0000 | 0.0000 | 1.0783 | 0.0262 | 0.0000 | 0.0000 | 0.0000 | 0.0000 |
| Southwest | 0.0000 | 0.0000 | 1.6714 | 1.1543 | 0.0000 | 1.3783 | 0.0000 | 0.0000 |
| Northwest | 1.2098 | 1.5462 | 1.5117 | 1.2458 | 1.0594 | 1.2385 | 1.3356 | 0.0000 |

**Table 7. The matching degree of embodied carbon trade and value-added trade among the eight regions in 2015.**

| Region | Northeast | North coast | East coast | South coast | Middle Yellow River | Middle Yangtze River | Southwest | Northwest |
|---|---|---|---|---|---|---|---|---|
| Northeast | 0.0000 | 0.0376 | 0.0316 | 0.0208 | 0.0000 | 0.0354 | 0.0214 | 0.0001 |
| North coast | 0.0000 | 0.0000 | 1.3648 | 0.0188 | 0.0000 | 0.0215 | 0.0217 | 0.0000 |
| East coast | 0.0000 | 0.0000 | 0.0000 | 0.0077 | 0.0000 | 0.0000* | 0.0036 | 0.0000 |
| South coast | 0.0000 | 0.0000 | 0.0000 | 0.0000 | 0.0000 | 0.0000 | 0.0000 | 0.0000 |
| Middle Yellow River | 1.2572 | 0.3671 | 0.0924 | 0.0392 | 0.0000 | 0.1040 | 0.0692 | 0.0000 |
| Middle Yangtze River | 0.0000 | 0.0000 | 0.0000 | 0.0055 | 0.0000 | 0.0000 | 0.0000 | 0.0000 |
| Southwest | 0.0000 | 0.0000 | 0.0000 | 0.0199 | 0.0000 | 1.3477 | 0.0000 | 0.0000 |
| Northwest | 0.0000 | 1.3648 | 1.4338 | 0.0746 | 1.0965 | 1.0000 | 1.3042 | 0.0000 |

Table 6, the value of the direction of transfer from the middle reaches of the Yellow River to the southwest region is 1, and in Table 7, the value of the direction of transfer from the northwest area to the middle reaches of the Yangtze River is 1. This is the maximum value among the set of data calculated according to Eq (11) $\frac{\bar{DN}^{rs}}{\bar{W}^{rs}}$. (4) Except for the values marked with $^*$, the matching values of embodied carbon trade and value-added trade between any two regions are only one value greater than 0. For example, in Table 6, the value of northeast to east coast outward direction is 0.1398, but the value of northeast from east coast inward direction is 0. This is because the trade relationship between any two regions either has embodied carbon net transfer out or embodied carbon net transfer in, and only when the net embodied carbon transfer from area r to area s is greater than 0, $\bar{DN}^{rs}$ is greater than 0, otherwise it is equal to 0. This is because a net shift of embodied carbon from region r to region s or a net shift of embodied carbon to region s is shown. A value greater than 0 in Tables 6 and 7 indicates a net shift of embodied carbon from region r to region s (including the 0 marked with $^*$), and a value equal to 0 indicates a net shift of embodied carbon from area r to area s (except for the 0 marked with $^*$).

The values between 0 and 1 in Tables 6 and 7 (including the 0 marked with $^*$) indicate a net shift of embodied carbon from area r to area s, as well as a net transfer of added value. For example, a net transfer of embodied carbon and value added is shown from the northeast region to the eastern coast, southern coast, middle Yangtze River and southwest region in 2012, and a net transfer of embodied carbon and value added from the northeast region to the eastern coast, southern coast, middle Yangtze River, southwest region and northwest area in 2015.

Values greater than 1 in Tables 6 and 7 indicate a net transfer of embodied carbon from region r to region s, and a net transfer of value added from region s. For instance, in 2012, the Northwest region had a net transfer of embodied carbon to the Northeast, North Coast, East Coast, South Coast, Middle Yellow River, Middle Yangtze River, and Southwest regions, but a net shift of value added from the above regions is shown. In 2015, the Northwest region had a net shift of embodied carbon to the North Coast, East Coast, Middle Yellow River, and Southwest regions, but a net shift of value added from the above areas is shown.

Further, this paper uses heat maps to analyze the degree of matching between embodied carbon trade and value-added trade among 31 Chinese provinces in 2012 and 2015, as shown in Figs 3 and 4.

From Figs 3 and 4, it can be seen that: (1)Since there is no embodied carbon trade and value-added trade between itself and itself, the diagonal elements are all 0. (2) Some elements with CI values close to 0 are also labeled in gray and white in the figures, so the number of gray-colored grids in Figs 3 and 4 is greater than the sum of the grids labeled in other colors. Compared with Fig 3, the number of gray cells is higher in Fig 4 because there is a larger gap between the maximum and minimum values in the corresponding data sets in Fig 4, and thus

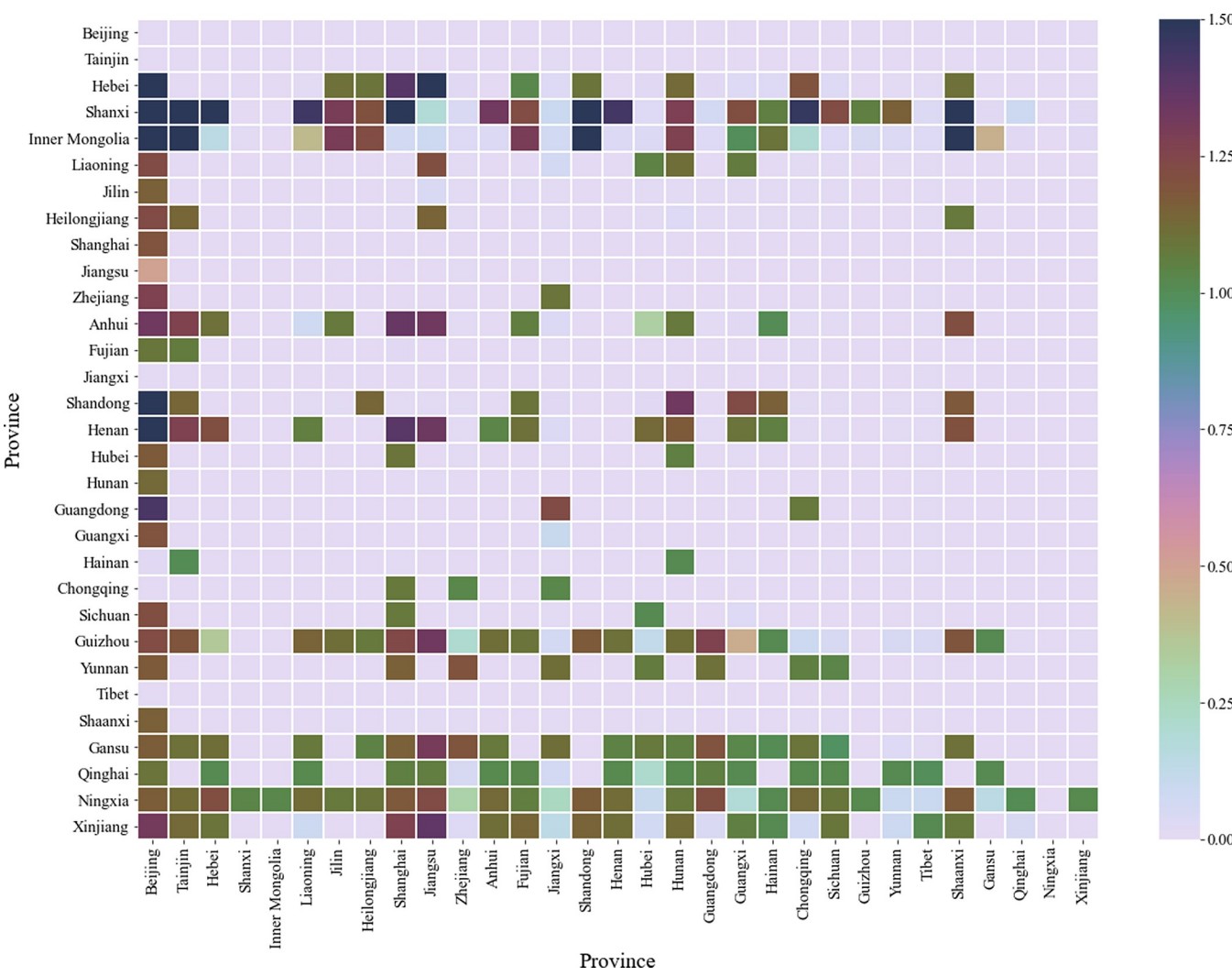

**Fig 3. The matching degree of embodied carbon trade and value-added trade among 31 provinces in China in 2012.**

more elements with CI values close to zero. (3) At the province level, a mismatch between embodied carbon trade and value-added trade is shown, for example, in Fig 3, there is a net embodied carbon transfer from Hebei to Beijing, but a net added value transfer from Hebei to Beijing; in Fig 4, there is a net embodied carbon transfer from Xinjiang to Beijing, but a net added value transfer from Xinjiang to Beijing.

## Further discussion

The input-output table of China (provinces)compiled by the national (provincial) bureau of statistics of China is usually the import competitive input-output table, while the multi-regional input-output table of China compiled in CEADs database is the import non-competitive input-output table. In the import competitive input-output table, the intermediate inputs consumed by each production departments do not distinguish between domestic production and import. It is assumed that they can be completely replaced, which means that the import column vector can be inserted directly in the third quadrant of the input-output table. While, the intermediate inputs in the noncompetitive input-output table are divided into two parts,

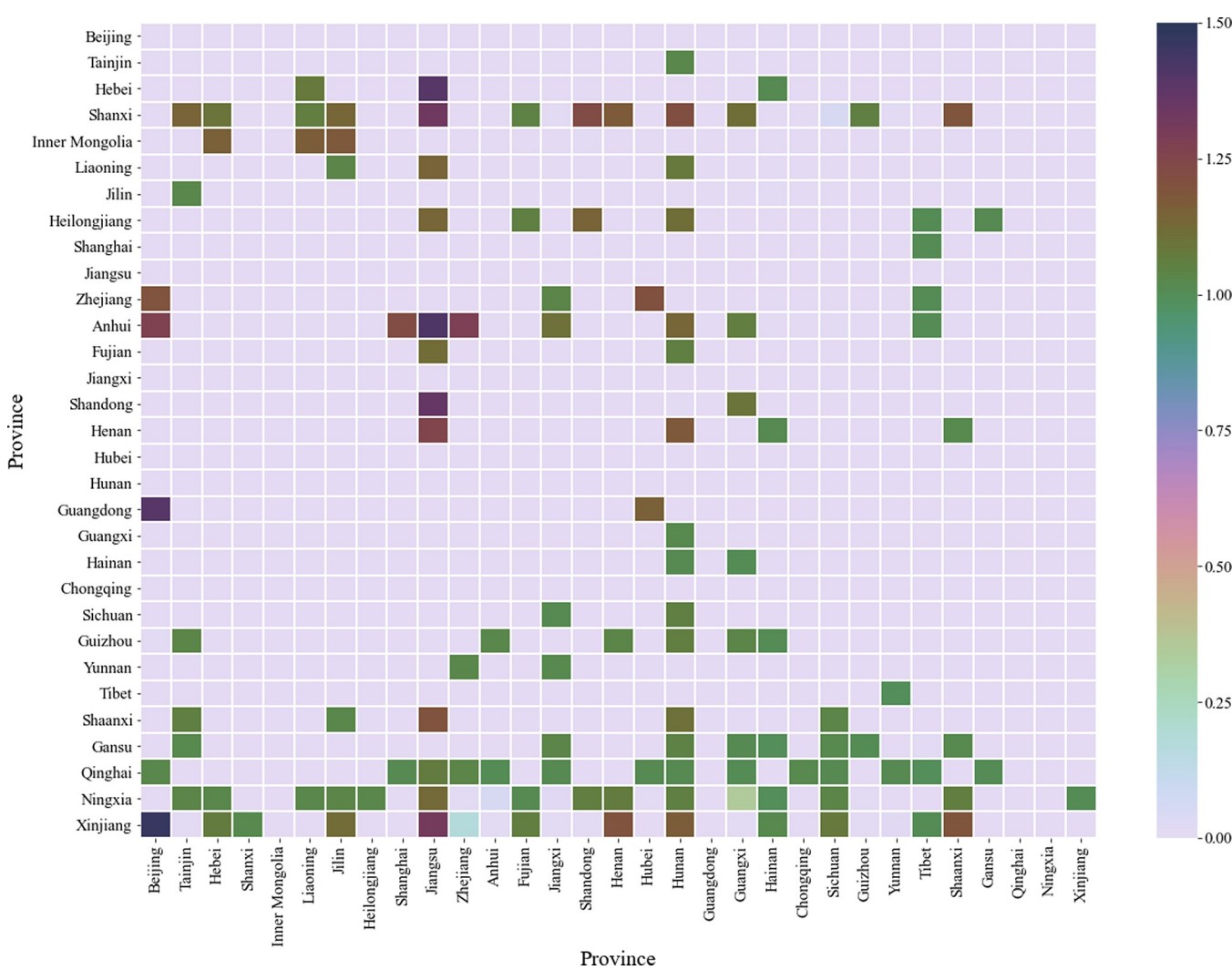

**Fig 4. The matching degree of embodied carbon trade and value-added trade among31 provinces in China in 2015.**

namely the intermediate input of domestic production and the intermediate input of imports, which reflects the incomplete substitution of the two. In addition, the import column vector also exists in the final demand part. Since the import non-competitive input-output table is adopted in this paper, it is not necessary to mention the impact of imported products. The related literatures on the import competitive and non-competitive input-output tables can be found in Su and Ang [23], and Su et al. [24].

The present paper is mainly concerned with the matching degree of embodied carbon trade and value-added trade among Chinese provinces (regions).Only the matching degree of domestic trade is considered, and the matching degree of international trade is not involved. Furthermore, the impact of international trade is not embedded in the consideration of domestic trade [9, 25, 26]. If the domestic trade and international trade are considered comprehensively, the matching degree between the embodied carbon trade and value-added trade of each province (region) may be different from the calculation results of this paper. Moreover, it is also the further research direction of this paper to study the driving factors of the matching

degree of embodied carbon trade and value-added trade among Chinese provinces (regions) combining with the structural decomposition analysis [27, 28].

It should be pointed out that, according to the multi-regional input-output model constructed in this paper, one can get the embodied carbon trade volume and the value-added trade volume between various provinces and industries (The data for each year isa930 × 31 order matrix, where, 30and 31 represent the number of industries and provinces, respectively. In this study, the input-output tables are merged in industries according to the carbon emission data, which contain 42 industries in the CEADs database).Based on the calculated embodied carbon trade volume and value-added trade volume of each province and industry in 2012 and 2015, it can better explain the results in Table 5 from the perspective of trade structure. Due to the large amount of data, it only reports the main results in this paper and does not explain the results in Table 5 from the perspective of trade structure, which is the inadequacy of this paper.

## Conclusions

This paper accounts for the embodied carbon trade and value-added trade among eight regions and 31 provinces in China using the inter-regional input-output tables and carbon emission data in China in 2012 and 2015, and investigates the matching degree of embodied carbon trade and value-added trade on this basis. The results of the study found that (1) in 2012 and 2015, the embodied carbon transaction between the middle Yellow River region and other areas was the largest, and the embodied carbon trade between the northwest region and other regions was the smallest. In 2012 and 2015, the value-added transaction between the eastern coastal region and other areas was the largest, and the value-added trade between the northwest region and other regions was the smallest.(2) A mismatch is shown between embodied carbon trade and value-added trade among regions, for example, in 2012, the northwest region had net embodied carbon transfer out to the north coast, but there is net value-added transfer in from the north coast. (3) In 2012 and 2015, Inner Mongolia had the largest net transfer out of embodied carbon, but in 2012 Zhejiang Province had the greatest net transfer in of embodied carbon, and in 2015 Guangdong Province had the greatest net transfer in of embodied carbon. In 2012 and 2015, Guangdong Province had the largest net transfer in of value added, but in 2012 Beijing had the largest net transfer out of value added, and in 2015 Jiangsu Province had the largest net transfer out of value added. (4) At the provincial level, there was also a mismatch between embodied carbon trade and value-added trade, for example, there was a net embodied carbon transfer out from Hebei to Beijing in 2012, but a net added value transfer in from Beijing in Hebei; there was a net embodied carbon transfer out from Xinjiang to Beijing in 2015, but a net added value transfer in from Beijing to Xinjiang.

## Supporting information

**S1 File. MRIO2012.**
(XLSX)

**S2 File. Province sectoral CO2 emissions 2015.**
(XLSX)

## Author Contributions

**Data curation:** Fengying Lu, Guangyao Deng.

**Funding acquisition:** Guangyao Deng.

**Methodology:** Guangyao Deng.

**Supervision:** Guangyao Deng.

**Writing – original draft:** Fengying Lu, Guangyao Deng.

**Writing – review & editing:** Xia Li.

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
