## [Decision Letter · Decision Letter 0]

25 Jul 2022

PONE-D-21-40527

Matching Degree of Embodied Carbon Trade and Value-added Trade among Chinese Provinces (Regions)

PLOS ONE

Dear Dr. Li,

Thank you for submitting your manuscript to PLOS ONE. After careful consideration, we feel that it has merit but does not fully meet PLOS ONE’s publication criteria as it currently stands. Therefore, we invite you to submit a revised version of the manuscript that addresses the points raised during the review process.

We look forward to receiving your revised manuscript.

Kind regards,

Xingwei Li, Ph.D.

Academic Editor

PLOS ONE

Journal Requirements:

2. Please note that PLOS ONE is designed to communicate original research and research methods, and we do not consider policy papers. As such, we ask you to remove policy discussion and recommendations made in your Conclusions section and ensure that statements made in your conclusions are supported by the data in your article.

5. Thank you for stating the following financial disclosure: "This work was supported by the Natural Science Foundation of China under Grant#71704070（GD）; Outstanding Youth Fund of Gansu Province#20JR5RA206（GD）; Gansu Provincial Higher Education Research Project#2020A-058（GD）; Double First-class Key Scientific Research Project of Gansu Provincial Department of Education # GSSYLXM-06（GD）；Key scientific research project of Silk Road Economic Research Institute of Lanzhou University of Finance and Economics # JYYZ202102（GD）; Program of Lanzhou University of Finance and Economics under Grant#Lzufe2021B-002（GD）.The funders had no role in study design, data collection and analysis, decision to publish, or preparation of the manuscript."

We note that one or more of the authors is affiliated with the funding organization, indicating the funder may have had some role in the design, data collection, analysis or preparation of your manuscript for publication; in other words, the funder played an indirect role through the participation of the co-authors. If the funding organization did not play a role in the study design, data collection and analysis, decision to publish, or preparation of the manuscript and only provided financial support in the form of authors' salaries and/or research materials, please do the following:

a. Review your statements relating to the author contributions, and ensure you have specifically and accurately indicated the role(s) that these authors had in your study. These amendments should be made in the online form.

b. Confirm in your cover letter that you agree with the following statement, and we will change the online submission form on your behalf: 

“The funder provided support in the form of salaries for authors [insert relevant initials], but did not have any additional role in the study design, data collection and analysis, decision to publish, or preparation of the manuscript. The specific roles of these authors are articulated in the ‘author contributions’ section.

6. Your abstract cannot contain citations. Please only include citations in the body text of the manuscript, and ensure that they remain in ascending numerical order on first mention.

7. Please upload a new copy of Figures 1-4 as the detail is not clear. Please follow the link for more information: https://blogs.plos.org/plos/2019/06/looking-good-tips-for-creating-your-plos-figures-graphics/" https://blogs.plos.org/plos/2019/06/looking-good-tips-for-creating-your-plos-figures-graphics/

Additional Editor Comments:

As mentioned by the reviewers, this manuscript must have undergone significant revision. Please carefully revise this manuscript based on the comments of the reviewers and respond point by point.

Reviewers' comments:

Reviewer's Responses to Questions

**Comments to the Author**

1. Is the manuscript technically sound, and do the data support the conclusions?

Reviewer #1: Yes

Reviewer #2: No

Reviewer #3: Yes

2. Has the statistical analysis been performed appropriately and rigorously? 

Reviewer #1: Yes

Reviewer #2: Yes

Reviewer #3: N/A

3. Have the authors made all data underlying the findings in their manuscript fully available?

Reviewer #1: Yes

Reviewer #2: No

Reviewer #3: No

4. Is the manuscript presented in an intelligible fashion and written in standard English?

Reviewer #1: Yes

Reviewer #2: Yes

Reviewer #3: Yes

5. Review Comments to the Author

Reviewer #1: This paper has discussed an interesting topic of matching between the transfer of embodied carbon emissions and value-added in China’s domestic setting. It is a research with important policy implications for carbon emission mitigation policy formulation. In addition to conventional input-output analysis, the authors went one step further to include a matching degree indicator to quantitatively compare the extent of mismatches between carbon and value-added trades.

However, I feel that the rationale of matching degree indicator can be much improved. Specifically, the reasons of parameter setting and algorithm design are not well explained for eq (11) and eq (12). Although the algorithm is directly taken from Chen et al., it will much greatly improve the readability if the authors can explain why the authors have designed an algorithm as such and what are the physical implications of each equations and variables. I also feel that the last paragraph on page 19 and first paragraph on page 20 are essentially explanation for the algorithm design. They will be better associated with the explanation of eq (11) and (12) in the Methodology section.

In addition, as the authors have clearly declared on page 6 that provinces are aggregated into eight regions in accordance with document by the State Council, why are the heat maps in fig. 3 and fig. 4 not following the eight regions specification? Please state the reasoning of your choice.

Lastly, the authors seem to be stating how this research is different from other literatures in Further Discussion. However, I feel the differences should be integrated into literature review in Introduction. Discussion section should cover limitations and improvements instead.

Reviewer #2: The article intends to estimate the embodied carbon trade and value-added trade within China, between its regions and provinces. Furthermore, the authors aim to reveal spatial disparities between these two flows in 2012 and 2015 with the help of MRIO-t. Unfortunately, though, several conceptual weaknesses of the analysis harm the robustness and the potential impacts on the field.

First, the authors did not justify their starting point. The statement "In order to better clarify the carbon emission reduction responsibilities of provinces (regions), it is necessary to analyze the degree of matching between embodied carbon trade and value-added trade by combining the calculation results of them" on the 2nd page is not enough to introduce the research question. Is a high degree of matching something to achieve, or contrary, a low degree of matching? If an environmental policy targeted to equalize embodied carbon trade among regions, potential resource savings would be lost, for instance, in agricultural production (see work by Carole Dalin). Without any consideration in this regard, one cannot evaluate the study results, and so the authors themselves were unable to implement them.

Second, the population and the economic performance of the regions (provinces) should be controlled. The authors should have used per capita and GDP values of carbon emissions and trade instead of gross values.

Third, the results are poorly explained concerning their meaning (see the first issue) and socio-economic context. For example, the discussion on the geographical distance on the top of page 9 is not supported by any data; significant differences in 'net transfer out' in Table 5 comparing the two analyzed years are also not addressed. The reader is not provided with any information about the economic and trade structure at the regional level, the composition of trade and embodied carbon flows, etc. That is exactly why one writes academic articles; to reveal and explain a phenomenon.

Merits of the manuscript do not reach the publication criteria of the journal.

Reviewer #3: This paper analyzed the embodied carbon and value-added in Chinese regional trade and constructed the associated matching index for year 2012 and 2015. Some interesting results have been presented in the paper. Below are the comments for the authors to improve the paper:

[1.Introduction]

Literature review is not sufficient. The authors are suggested to have a separate section to review the studies done in the literature, highlight the gaps and unique contributions of this paper to the literature. Particularly, the relationship between embodied carbon emissions and embodied value added has been studied using the aggregate embodied intensity (AEI) concept proposed by Su and Ang (2017; Energy Economics 65, 137-147). The AEI indicator can be defined at the aggregation, regional, final demand, sectoral and transmission layers. See, for example, the multi-region AEI studies in Wang et al. (2020; Energy Economics 85, 104568), Su et al. (2021; Journal of Cleaner Production 313, 127894) and Wang et al. (2022; Technological Forecasting & Social Change 177, 121546) for China’s regions.

[2.Methods]

When formulating the national or multi-regional IO analysis, it is important to clarify the imports assumptions used in the analysis. See the first paper by Su and Ang (2013; Energy Policy 56, 83-87) discussing the impacts of imports assumptions in embodied emission estimates, and recent study by Su et al. (2022; Energy Economics 107, 105875) on the impacts of imports data treatments. The authors are suggested to add more discussions on imports assumptions as China usually compiled its IO tables using the competitive-imports assumption.

When using only China’s MRIO table, the embodied flows only capture China’s interregional feedback effects, not including the international feedback effect (Su and Ang, 2011; Ecological Economics 71, 42-53). See, for example, the hybrid model in Su and Ang (2014; Applied Energy 114, 377-384) and embedded model in Su et al. (2021; Journal of Cleaner Production 313, 127894). The calculation of the matching index could be different when international feedback effects are captured. The authors are suggested to add more discussions in this sector and the limitations (or future research) of the study in the last section.

Eq (11): How to do the calculation when W^rs=0 or W^rs becomes very small, e.g. 10^(-30)? Robustness of the matching index formulation should be discussed.

Data: CEADS only compiled China’s regional carbon emissions for 30 regions, not including Tibet (not “Xizang” in the table). Where did the authors get the energy/emissions data for Tibet in 2012 and 2015?

[3.Results and Discussions]

The authors presented the results in the eight aggregated regions, which losing some critical information at the detailed provincial level. The authors are suggested to include all the 31 regions’ results (e.g. embodied emissions and value added in interregional trade) in the discussions or at least provide the detailed results in table in the Appendix.

Table 5: Are the embodied value added figures given for 2012 and 2015 comparable if not giving them in the fixed year price?

The authors are also suggested co compare their results (e.g. embodied emissions) with those reported in the literature using the similar MRIO dataset for China, including the regional AEI studies for China.

[Conclusions and implications]

The authors are suggested to add more discussions on the findings on the mismatching index, which is the major contribution in the paper, for policy makings. What are the possible ways to improve the inequality or mismatching for future regional developments?

The authors may consider using the structural decomposition analysis (SDA) to investigate the drivers of the matching index in future research. See the review of SDA studies on energy and emissions in Su and Ang (2012; Energy Economics 34 (1), 177-188) and Wang et al. (2017; Energy Policy 107, 585-599).

6. PLOS authors have the option to publish the peer review history of their article (what does this mean?). If published, this will include your full peer review and any attached files.

Reviewer #1: No

Reviewer #2: No

Reviewer #3: No

---

## [Decision Letter · Decision Letter 1]

21 Oct 2022

Matching Degree of Embodied Carbon Trade and Value-added Trade among Chinese Provinces (Regions)

PONE-D-21-40527R1

Dear Dr. Deng,

We’re pleased to inform you that your manuscript has been judged scientifically suitable for publication and will be formally accepted for publication once it meets all outstanding technical requirements.

Kind regards,

Xingwei Li, Ph.D.

Academic Editor

PLOS ONE

Additional Editor Comments (optional):

Reviewers' comments:

Reviewer's Responses to Questions

**Comments to the Author**

1. If the authors have adequately addressed your comments raised in a previous round of review and you feel that this manuscript is now acceptable for publication, you may indicate that here to bypass the “Comments to the Author” section, enter your conflict of interest statement in the “Confidential to Editor” section, and submit your "Accept" recommendation.

Reviewer #3: All comments have been addressed

Reviewer #4: All comments have been addressed

2. Is the manuscript technically sound, and do the data support the conclusions?

Reviewer #3: Yes

Reviewer #4: Yes

3. Has the statistical analysis been performed appropriately and rigorously? 

Reviewer #3: N/A

Reviewer #4: Yes

4. Have the authors made all data underlying the findings in their manuscript fully available?

Reviewer #3: Yes

Reviewer #4: Yes

5. Is the manuscript presented in an intelligible fashion and written in standard English?

Reviewer #3: Yes

Reviewer #4: Yes

6. Review Comments to the Author

Reviewer #3: The authors have clearly addressed the comments in the revision. I would suggest accepting it for publication in the journal.

Reviewer #4: (No Response)

7. PLOS authors have the option to publish the peer review history of their article (what does this mean?). If published, this will include your full peer review and any attached files.

Reviewer #3: No

Reviewer #4: No

---

## [Editor Report · Acceptance letter]

28 Oct 2022

PONE-D-21-40527R1 

Matching degree of embodied carbon trade and value-added trade among Chinese provinces (regions) 

Dear Dr. Deng:

I'm pleased to inform you that your manuscript has been deemed suitable for publication in PLOS ONE. Congratulations! Your manuscript is now with our production department. 

Kind regards, 

on behalf of

Prof. Dr. Xingwei Li 

Academic Editor

PLOS ONE